# Humanoid Parkour Learning

**Ziwen Zhuang[123], Shenzhe Yao[12], Hang Zhao[13]**
[1]Shanghai Qi Zhi Institute, [2]ShanghaiTech University, [3]Tsinghua University
Project Website: https://humanoid4parkour.github.io

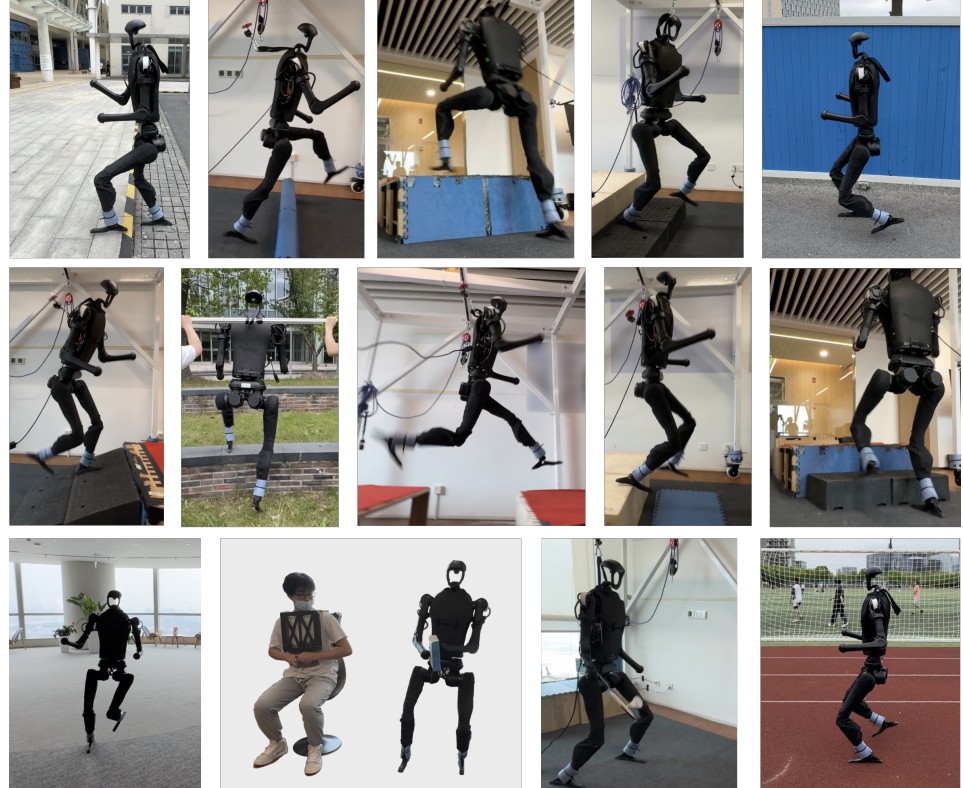

Figure 1: We present a single vision-based end-to-end whole-body-control parkour policy for humanoid robots that can jump on 0.42m platforms, leap over hurdles, 0.8m gaps, and overcome various terrains.

**Abstract:** Parkour is a grand challenge for legged locomotion, even for quadruped robots, requiring active perception and various maneuvers to overcome multiple challenging obstacles. Existing methods for humanoid locomotion either optimize a trajectory for a single parkour track or train a reinforcement learning policy only to walk with a significant amount of motion references. In this work, we propose a framework for learning an end-to-end vision-based whole-body-control parkour policy for humanoid robots that overcomes multiple parkour skills without any motion prior. Using the parkour policy, the humanoid robot can jump on a 0.42m platform, leap over hurdles, 0.8m gaps, and much more. It can also run at 1.8m/s in the wild and walk robustly on different terrains. We test our policy in indoor and outdoor environments to demonstrate that it can autonomously select parkour skills while following the rotation command of the joystick. We override the arm actions and show that this framework can easily transfer to humanoid mobile manipulation tasks. Videos can be found at project website.[1]

**Keywords:** Humanoid Agile Locomotion, Visuomotor Control, Sim-to-Real Transfer

---

[1]Correspondence Email: hangzhao@mail.tsinghua.edu.cn

8th Conference on Robot Learning (CoRL 2024), Munich, Germany.

# 1 Introduction

Athletic Intelligence is an important field of research when developing an embodied intelligence system. To perform daily human tasks or replace humans in some dangerous tasks, we need to develop legged robots, quadrupeds, or humanoids, that can overcome all human-capable terrains. Parkour is an exemplary task of athletic intelligence, requiring robots to move fast and deal with various obstacles, such as jumping up to a high platform, leaping through a long gap, and jumping over a high hurdle. These tasks require a unified system to perceive and memorize its surroundings [1, 2], autonomously make decisions to select proper skills, and execute challenging tasks with its powerful hardware [3, 4, 5]. In the context of robot learning, building a general parkour system that runs fully onboard without external support and without limitations to the non-critical properties of the terrain[2] is the grand challenge to explore the limit of an athletic intelligence system.

Recent progress in humanoid robots starts with Boston Dynamics Atlas robots [6], which perform stunning parkour acts [7] in their lab environment. However, the agile ability comes from massive engineering efforts to plan all the parkour acts offline and is most likely limited to that one obstacle order. Also, the high cost of hardware construction and maintenance restricts most humanoid skills only in simulation [8, 9]. Recent research efforts on learning-based quadruped and bipedal robot locomotion have demonstrated impressive robustness in walking [10, 11, 12], walking up and down stairs [13, 14, 15] and overcoming challenging terrains [4, 3, 16, 17, 18, 19, 20]. However, learning-based methods for humanoid robots are still limited to walking on the plane [21, 22, 23, 24, 25] where the legs cannot produce task-specific motions without any motion prior. It is great timing to explore a general way of building locomotion models with less motion prior so that humanoid robots can fully exploit the embodiment and perform tasks beyond plane locomotion.

Based on recent progress in humanoid locomotion, more agile locomotion for humanoids has several challenges. First, diverse skills for a single locomotion network are challenging. Previous methods in quadruped rely on specific engineering for each locomotion task. Humanoid locomotion based on learning methods requires reward terms to encourage foot-raising and prevent drag on the floor. These implementations significantly limit the legs' motion and keep them away from generating more diverse skills. Second, mimicking animals or humans lacks egocentric perception, and a huge amount of data for different types of terrain and embodiments is needed. This significantly restricts the scalability of these methods. Third, proprioception and exteroception take time to process. The system lagging must be strictly controlled so that the entire robotic system can act on time. Lastly, parkour skills exploit the maximum actuator capacity. Using electric motors with batteries brings significant challenges to the power throughput. Proactive action smoothness and restrictions are needed to mitigate the potential damage to the humanoid robot hardware.

This paper introduces a unified humanoid parkour learning system for at least 10 types of human-capable terrains, such as jumping up, jumping down, leaping over gaps, and jumping over hurdles. Our reinforcement learning system proves that fractal noise in terrain, which is frequently used in quadruped robots, trains a deployable humanoid locomotion skill without any motion reference or reward term to encourage foot raising. Our training objective is simple enough to train multiple agile humanoid locomotion skills in a unified manner while being able to deploy to the real humanoid robot with zero-shot sim-to-real transfer. Because of the straight parkour track, we train our parkour policy from a pretrained plane locomotion policy, so that the policy will respond to the turning command even if the locomotion command tells the robot to walk along the straight track. We then use DAgger [26, 27] with 4-GPU acceleration to distill a vision-based parkour policy that can be deployed on the real humanoid robot with only onboard computation, sensing, and power support.

---

[2]Non-critical properties example: the texture of a jumping platform, the depth of a gap for leaping, and the order of a series of obstacles

## 2 Related Work

**Legged Locomotion** Legged locomotion research starts with model-based control [28, 29, 30, 31]. Either quadruped robots or bipedal robots require heavy mechanical design and dynamics modeling to implement the control algorithm [28, 29, 30, 31, 30]. These methods are all aimed at walking in the plane and as robust as possible. Till the release of the general physical simulator[32, 33], reinforcement learning algorithms can be applied to legged locomotion tasks [34, 35, 36, 37, 19, 10, 38, 39]. But due to lack of exteroception, most of the learning algorithms are limited to low-speed locomotion [19, 40]. To perceive the terrain around the robot, typical methods use vision and depth sensors to estimate the robot state [41, 42, 43, 44] and response to different terrain [15, 45, 35, 3, 4, 46, 20]. Classical methods involves state-estimation [41, 42, 43, 44], traversability mapping [47, 48, 49, 50], and then plan their foothold [51, 18, 52, 45]. Learning based end-to-end system reacts to different terrains by depth vision [35, 53, 54], recovering elevation maps [13, 55, 56] or learned neural spaces [46, 57]. But only recently, locomotion on quadruped robot has demonstrated extreme agile locomotion using depth-sensing pipeline [3, 4, 58] .

**Humanoid Locomotion with Learning Algorithm** With the rapid development of reinforcement learning research and many zero-shot sim-to-real transfer algorithms applied on legged robots, bipedal robots such as Cassie [59] have already been deployed with various reinforcement learning algorithms [14, 60, 61, 62, 20]. However, due to the low inertia of the body and the start of the humanoid industry, little research on humanoid learning algorithms has been done. Using reinforcement learning and transformer architecture, humanoid robots can walk smoothly in the wild [22]. Arms movement not affecting the balancing and locomotion of the legs has also been verified [21, 23]. However, these methods still require significant reward engineering, either reference trajectory [22, 63] or encouraging feet raising [21, 23], to achieve possible sim-to-real robot behavior. In this work, we propose using fractal noise [11] on the terrain to encourage foot raising while leaving sufficient freedom to train downstream whole-body control tasks.

## 3 Training Method and Robot System

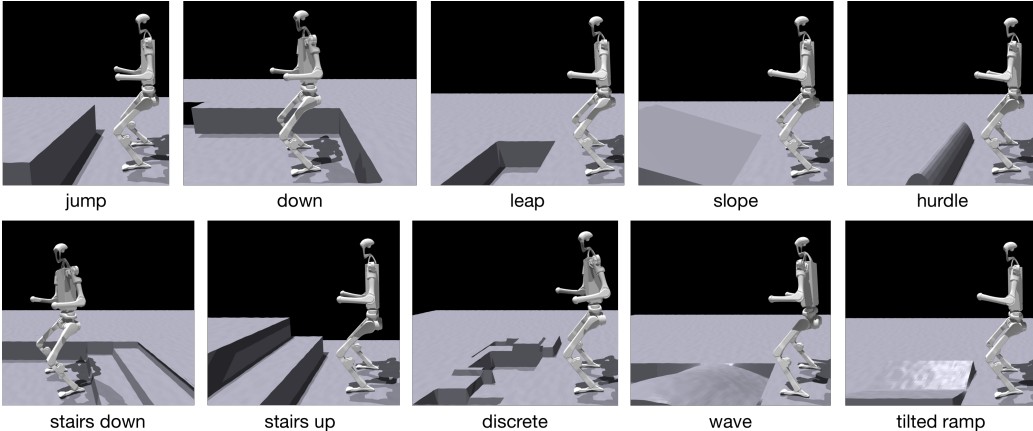

Figure 2: We design 10 different types of terrain with controllable difficulty. By training on all these terrains, the oracle policy is able to react to almost all human-capable terrain.

In this section, we aim to describe our implementation that trains a zero-shot end-to-end policy that uses onboard depth vision and robot proprioception to autonomously do parkour tasks when encountering various obstacles. Since rendering depth image in isaacgym [33] costs too much time, recent sim-to-real methods use prebuilt terrain information to train an oracle policy and then distill a deployable policy by switching the exteroception encoder. We divide the training pipeline into 3 stages. We first train a walking policy that follows locomotion commands for walking forward, walking sideways, and turning. In order to train the walking policy with proper gait in the real

world, we add fractal noise to the height field of the terrain [11], so that the robots learn to raise their foot when moving around. Then, we train that parkour policy with all parkour skills using an auto-curriculum mechanism. We use scandots as a means of perceiving the terrain because the height field is stored in GPU memory when the terrain is being built. We implement 10 different types of terrain as shown in Figure 2, so that the oracle policy will see as much terrain as possible and generalize to almost all human-capable terrain. We also add virtual obstacles that encourage the policy to keep away from dangerous behavior [3]. Then, we simulate the noise pattern of the Intel RealSense D435i stereo depth camera and distill a parkour policy using DAgger [26, 27]. Lastly in this section, we describe how we deploy our parkour policy on the real robot.

## 3.1 Training Forward Parkour from Planar Walking

To build a parkour policy that also follows the command of high-level navigation, the robot must be equipped with a turning ability. Since we build the the parkour track in a straight line so that the robot can autonomously keep walking to the next track, the turning ability must be pretrained. Considering perceiving the terrain using depth sensing is quite costly due to the implementation of isaacgym [33], we use scandots to acquire terrain information, shown in the supplementary. Different from [3], we don't need to represent obstacles that are above the robot, so scandots are general enough.

We formulate the oracle policy using a Gate Recurrent Unit (GRU) [64] and an MLP as the backbone of the actor. We use an MLP to encode a $s_t^{\text{scandot}} \in \mathbb{R}^{11 \times 19}$ scandot observation and return a $s^{\text{embedding}} \in \mathbb{R}^{32}$ terrain embedding. We use a GRU connected by an MLP as the state estimator to predict the linear velocity $\hat{v}_t \in \mathbb{R}^3$ of the robot base using the onboard available proprioception $s_t^{\text{proprio}} \in \mathbb{R}^{43}$ (row, pitch, base angular velocities, positions and velocities of joints) and last action $a_{t-1} \in \mathbb{R}^{19}$. Then, the oracle policy uses terrain embedding $s^{\text{embedding}}$, estimated base velocity $\hat{v}_t$, and proprioception $s_t^{\text{proprio}}$ to predict an action $a_t$ as all of the target joints of the robot body.

We use PPO [65] to train our policy. Then, the value network uses the same architecture as the policy network, except using the ground-truth linear velocity from the simulation.

For better sim-to-real deployment, we use multiple domain randomization techniques on the robotics dynamics. We describe the details in the supplementary.

**Train a Walking Policy on Plane**  To train a general plane-walking policy that follows the x, y, yaw moving command, we used fractal noise [11] and multiple reward terms. The rewards and their weights are shown in the

| Term | Range |
|---|---|
| x velocity | [-0.8 m/s, 2.0 m/s] |
| y velocity | [-0.8 m/s, 0.8 m/s] |
| yaw velocity | [-1 rad/s, 1 rad/s] |

Table 1: Command ranges when training a plane-walking policy

supplementary. The reward function consists of mainly three parts: 1. Task rewards, including linear velocity tracking, angular velocity tracking, and minimizing total energy consumption; 2. Regularization rewards, including torque penalty, joint acceleration penalty, joint velocity penalty, and foot contact penalty; 3. Safety rewards, including unnecessary joint movements penalty, such as arm and waist, keeping both feet away from each other, and lower action rate. As Table 1 shows, we sample the moving command uniformly and train the walking policy.

**Auto Command to Walk along the Track**  To meet the needs of following navigation commands when walking in the plane and overcoming parkour obstacles, the walking policy is equipped with walking and turning ability. However, the parkour track is a straight line along the x-axis. Putting the robot facing +x direction and sending only forward command will lead to the loss of turning ability. We initialize the robot orientation uniformly in all directions and design an autonomous command mechanism to guide the robot to turn to the parkour track.

Suppose the goal position is in the direction of $\theta_{\text{goal}}$ of the robot and the robot orientation is $\theta$, the yaw command will be $v_{\text{yaw}}^{\text{cmd}} = \alpha_{\text{yaw}} * (\theta_{\text{goal}} - \theta)$, where $\alpha_{\text{yaw}}$ is a predefined ratio. If $\|\theta_{\text{goal}} - \theta\| >= \pi/2$, the forward-moving command will be set to zero. Otherwise, the forward-moving command will be set to a sampled value in Table 1. Then, the trained policy will keep following the straight parkour track while preserving the turn-by-command ability.

**Train an Oracle Parkour Policy**   To generate different terrain for massively parallel simulation, we put 10 types of parkour obstacles in a grid of 10 rows and 40 columns. Each column is a single type of obstacle, shown in Figure 2. Each row is an obstacle with different difficulty. For the $i$-th row, the critical parameters of the obstacle are scored by $(1 - i/10) * l_{\text{easy}} + i/10 * l_{\text{hard}}$. In each sub-terrain, the parkour track consists of a starting plane to initialize the robot and an obstacle block where the obstacle is built. When the sampled forward command is greater than zero, the robot will move to a more difficult terrain if the robot safely finishes the task and moves at least $3/4$ distance of the sub-terrain. The robot will move to an easier terrain if the robot fails at a distance smaller than $1/2$ of the sub-terrain. The ranges for the critical properties of each parkour obstacle are listed in the supplementary.

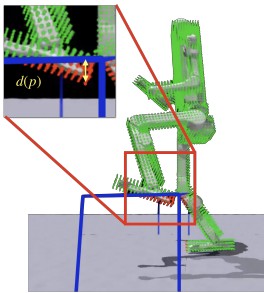

Figure 3: All the points are bound to each rigid body of the robot. The green points represent that part of the body does not penetrate the virtual obstacle. The red points represent that the body penetrates the virtual obstacle.

In order to generate a safer policy with action redundancy and more accurate action steps, we add two reward terms to train the parkour policy.

For several tasks such as leaping, the robot tends to exploit the edge, which is sensitive to motor error. We add virtual obstacle [3] in some of the terrains, e.g. leap and jump. We compute the penetration penalty and apply it during reward computation. As Figure 3 shows, we bind a mesh of points around each link of the robot body. The dark blue line draws the virtual obstacle, which the robot should avoid. The green points show the estimated volume does not penetrate any virtual obstacle. The red points show the estimated volume penetrating the virtual obstacle. For this hurdle obstacle, the penetration surface is on the top. Then, the penetration depth $d(p)$ for a given point is shown in Figure 3. We compute penetration penalty as an additional reward term by

$$r_{\text{penetrate}} = \alpha_{\text{penetrate}} * \sum_{p} (d(p) * \|v(p)\|), \tag{1}$$

where $p$ is the penetration depth of all red points bound on the robot bodies. $\alpha_{\text{penetrate}} = -5 \times 10^{-3}$ in our case.

For tasks such as stairs up and stairs down, reinforcement learning does not generate accurate enough action without explicit footstep guidance [66]. We assign the recommended stepping point to the middle of each stair. We ignore the stepping distance perpendicular to the forward direction. When the robot is walking on stairs, the offset between the recommended stepping position and the actual stepping position is $d = (d_x, d_y, d_z)$. We compute the footstep reward by

$$r_{\text{step}} = \alpha_{\text{step}} * (-\ln \|d_x\|), \tag{2}$$

where $\alpha_{\text{step}} = 6$ in our case.

## 3.2   Distill Visual Perception from Scandots

**Initialize Student Policy from Teacher Policy**   In order to fully utilize the pretrained oracle model, we replace the original scandots encoder with a randomly initialized CNN network, which encodes a $I_t \in \mathbb{R}^{48 \times 64}$ depth image into the terrain embedding $\hat{s}^{\text{embedding}} \in \mathbb{R}^{32}$. We initialize the weight of the GRU and MLP module of the student policy using the weight of the Oracle parkour policy. We keep using the well-trained linear velocity estimator and keep optimizing this module.

**Bridge the Depth Image Gap between Sim and Real**   While training the oracle parkour policy in Section 3.1 bridging the gap in physical and mechanical dynamics using various domain randomization techniques, the terrain information acquired by the oracle parkour policy is still lossless and unavailable in the real world. We address this gap by simulating the noise pattern of the depth image and applying built-in filters in the Intel RealSense API when acquiring depth images onboard. Shown in Figure 4, we apply depth clipping, multi-level Gaussian noise, and random artifacts to the raw rendered depth image. We apply depth clipping, hole filling, spatial smoothing, and temporal smoothing to the raw stereo-depth image in the real world.

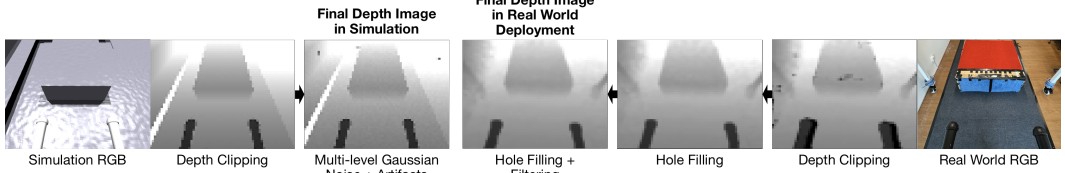

Figure 4: From left to right is the process of simulating noise in simulation. From right to left is the process of pre-processing the depth image in the real world.

**Multi-Processes Distillation Acceleration**    We observe a single Nvidia 3090 GPU is inefficient in rendering depth images in the IsaacGym simulator. We use 4 Nvidia 3090 GPUs in 4 processes to accelerate the DAgger [26, 27] distillation process. We use one process as the trainer and keep loading the student trajectories and teacher action labels. We use the other 3 processes as the collectors to collect the student trajectories and label them using the Oracle parkour policy as the teacher. The trainer and collectors pass the latest student policy and the labeled trajectories using a shared filesystem, which all processes can access. Since both student policy and Oracle parkour policy outputs action in the real number domain, we distill the student policy using the L1 norm between the student action and teacher action as the objective.

### 3.3  Zero-shot Sim-to-Real Deployment

We directly copy the distilled parkour policy model to the Unitree H1 robot and use the same network implementation during training. We run the visual encoder and the rest of the network on two separate Python processes on a 12-core Intel i7 CPU. These two processes communicate with each other and with the motors on the humanoid robot using Cyclone DDS as well as the ROS 2 platform. For the depth vision, we use RealSense D435i which is mounted on the head of the humanoid. Other details will be presented in the supplementary file.

## 4   Experiment Results

In this section, we first describe the setup for our humanoid robot and the experiment setup in simulation and in the real world. Then, we use three separate experiments to demonstrate the following questions: 1. Is using fractal noise terrain feasible compared to "feet air time" reward or motion reference to make the foot raise 2. Concurrent works use "feet air time" or motion prior [67] to make the walking policy able to walk up and down stairs without vision. Is the vision for challenging terrain necessary? 3. Previous Parkour [3, 4] uses either single-GPU distillation or distill a student policy from scratch. Why does our method require both initializing student policy from the oracle policy and using multi-GPU acceleration?

### 4.1   Robot and Experiment Setup

We use Unitree H1 and their official urdf model to train and test the policy in isaacgym [33] simulator. We evaluate different tasks in simulation by building 10 straight tracks. Each track corresponds to a single type of obstacle. Each track has 3 connected sub-tracks with 3 linearly increasing difficulties on the obstacle properties ranges in the supplementary.

For the real-world experiment, we use several coffee tables with a height of 0.42m and several wooden plyo boxes with a height of 0.4m (16 inches) for jumping up and down, as well as leaping. We use curb ramps for the slopes of 0.2m height and 0.5m length (0.38 rad). We use 0.2m flat ramps to build small stairs with wooden boxes. We put the curb ramps side by side to test the tilted ramp terrain. We use foam pads to build low hurdles with heights of 0.25m.

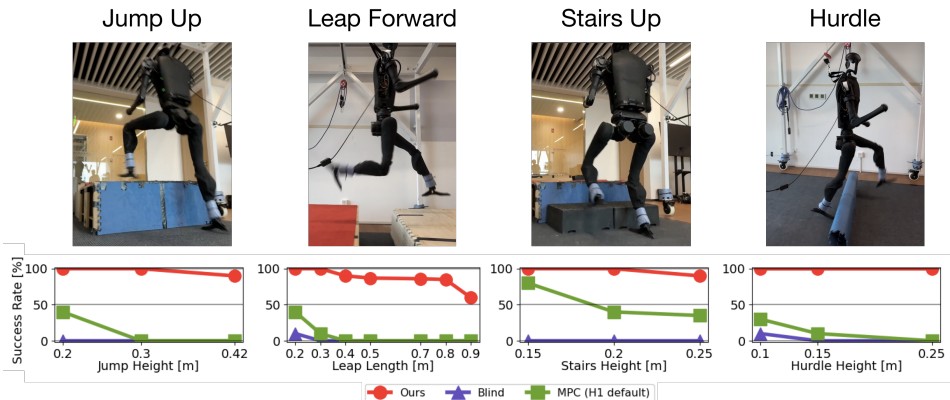

Figure 5: Real-world quantitative results. Our parkour policy achieves the best performance in the 4 difficult tasks compared with the blind walking policy and the built-in MPC controller in Unitree H1. We use the user interface in H1's controller to raise its foot height so that it can overcome some of the obstacles. We run 10 trails for each testing configuration.

## 4.2 Fractal Noise with No Motion Prior is Effective

We compare the success rate and average moving distance between the oracle policies trained with fractal noise terrain (ours) and with "feet airtime" [21, 23] reward terms in all 10 skills in simulation. We deploy 10 robots for each type of terrain with no fractal noise and put them in the easiest sub-track. Shown in Table 2, the success rate is computed from the number of robots that successfully overcome the terrain with maximum difficulty. The average moving distance is computed from the average distance before the robot falls or stops in a given type of terrain. Shown in Table 2, our method that trains on fractal noise terrain performs better than that using "feet air time" in the reward function, which further illustrates that the "feet air time" term conflicts with some of the locomotion tasks.

| | Success Rate (%) ↑ | | | | Average Distance (m) ↑ | | | |
|---|---|---|---|---|---|---|---|---|
| | Jump up | Leap | Stairs up | Hurdle | Jump up | Leap | Stairs up | Hurdle |
| feet_air_time | 80 | 50 | 90 | 20 | 9.81 | 13.4 | 13.6 | 12.1 |
| Fractal (Ours) | **90** | **80** | **100** | **100** | **14.3** | **14.0** | **14.4** | **14.4** |

Table 2: We compare the success rate and average moving distance in simulation. Each type of terrain is a connected 3 sub-track with linearly increasing difficulty. Each sub-track has a length of 4.8m, which makes the total length of a given terrain 14.4m. The success rates are averaged across 10 runs for each case. (More experiment results for the other obstacles are shown in the appendix.)

## 4.3 Onboard Vision is Crucial

We perform real-world quantitative experiments to measure the performance of our parkour policy. Due to the safety issue, we quantitatively test the success rate of the 4 most difficult tasks in the indoor environment. We change the difficulty by modifying the key properties of each obstacle, such as the height of the platform, the gap length for leaping, and the height of each stair. Since our parkour policy follows a directional command, we filter out the operator failure. Shown in Figure 5, we test our policy in the real-world lab environment. The robot can select the proper skill in challenging parkour tasks based on its onboard depth vision.

## 4.4 Distill from Expert Policy and Multi-Processing Acceleration is Critical

We compare the efficiency of using multi-GPU acceleration in terms of transitions experienced by the student policy network and the success rate after 24 hours of training. The student policy distilled from scratch still struggles to keep balance. The student policy distilled using one GPU reaches only half of the performance of the fully accelerated distillation. Details are described in the supplementary.

## 4.5 Observations in Locomotion Behavior

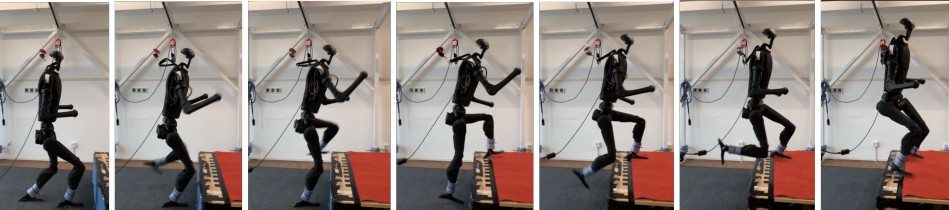

Figure 6: In this example of jumping up, the right arm of the humanoid robot emerges a large swinging pattern to keep balance.

We observe that under the arm regularization term of the reward function, the whole-body-control parkour policy is still able to manage arm movement. During jumping up, the parkour policy swings up the right arm to keep the arm away from the body while keeping balance during the agile maneuver.

## 4.6 Upper Limbs Control Does Not Lead to Policy Failure

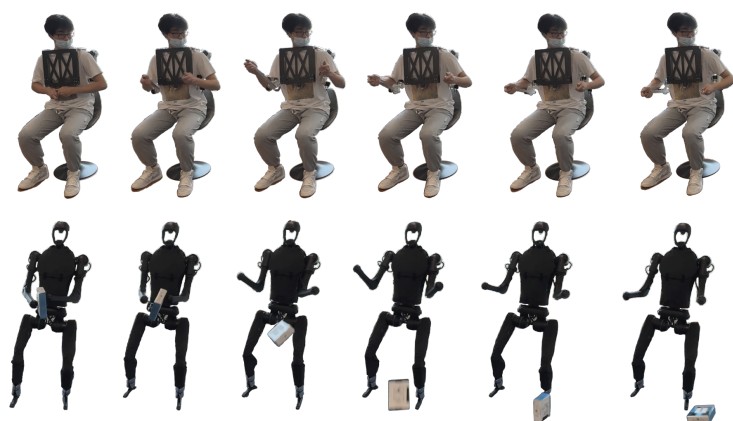

Figure 7: We use a teleoperation setup to override the action of the parkour policy. The robot can keep balance while the arms are waving around. To be noted that this is an out-of-distribution situation for the parkour policy.

Previous work in humanoid robots stated that overriding the arms action in a whole-body-controlled policy will significantly disrupt the locomotion stability. We aim to verify this statement. In the real world, we build a set of joint sensors that map human arm motion to Unitree H1 arm joints. Shown in Figure 7, the robot runs the parkour policy while the arm is teleoperated by a human operator. The robot can still keep balance while the arm is not acting under the policy's command.

## 5 Conclusion, Limitations and Future Directions

We present a parkour learning system that can be applied to humanoid robots, without complex motion reference. We propose using fractal noise and a two-stage training method to simplify the reward function and let the parkour policy follow the turning command even when the parkour track is straight. We test our method on Unitree H1 to show the effectiveness of our framework. However, our method relies on manually constructed terrain. It is difficult to generalize to unseen terrains without further training. Parkour is a whole-body-control task with extreme difficulty. We show that the parkour policy is robust to arm-action override, which can be used in manipulation tasks. However, more difficult manipulation skills require further tuning to avoid disrupting the visual system. We will investigate how to utilize additional training to cooperate with manipulation skills while preserving the extreme locomotion ability.

**Acknowledgments**

Thanks to Qiao Sun, Derun Li, Shaoting Zhu, Xin Duan, Hao Fu, and Honglei Zhang for the video shoot. We thank Xuxin Cheng, Shaoting Zhu, Xin Duan, Derun Li, and Miao Zhuang for their help with experiments, valuable discussions, and support. This project is supported by Shanghai Qi Zhi Institute and outdoor videos are supported by ShanghaiTech University.

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

# A  Experiment Videos

We perform through real-world video analysis of our system. We perform indoor parkour tests and outdoor locomotion tests. Videos can be found at https://humanoid4parkour.github.io

# B  Domain Randomization Details

Table 3: Terms and Ranges for Domain Randomization

| | |
|---|---|
| Added Mass (kg) | [-1.0, 5.0] |
| Center of Mass x (m) | [-0.1, 0.1] |
| Center of Mass y (m) | [-0.15, 0.15] |
| Center of Mass z (m) | [-0.2, 0.2] |
| Friction | [-0.2, 2.0] |
| Motor Strength | [0.8, 1.2] |
| Proprioception Latency (s) | [0.005, 0.045] |
| Depth Field of View (degree) | [86, 90] |
| Depth Latency (s) | [0.06, 0.12] |
| Depth Camera Position x (m) | [0.1, 0.12] |
| Depth Camera Position y (m) | [-0.02, -0.015] |
| Depth Camera Position z (m) | [0.64, 0.7] |
| Depth Camera Row (rad) | [-0.1, 0.1] |
| Depth Camera Pitch (rad) | [0.77, 0.99] |
| Depth Camera Yaw (rad) | [-0.1, 0.1] |

We uniformly sample all attributes in Table 3 across all 4096 robots during reinforcement learning and distillation.

# C  Reinforcement Learning Training Details

Table 4: Reward terms specification

| Term | Expression | Weights |
|---|---|---|
| Linear Velocity Tracking | $\exp(-\|v - v^{\mathrm{cmd}}\|/0.25)$ | 1.0 |
| Angular Velocity Tracking | $\exp(-\|v_{\mathrm{yaw}} - v_{\mathrm{yaw}}^{\mathrm{cmd}}\|/0.25)$ | 1.5 |
| Orientation | $g_x^2 + g_y^2$ | -2. |
| Energy | $\sum_{j \in \mathrm{joints}} |\tau_j \dot{q}_j|^2$ | -2.5e-7 |
| DoF Velocity | $\sum_{j \in \mathrm{joints}} |\dot{q}_j|^2$ | -1e-4 |
| DoF Acceleration | $\sum_{j \in \mathrm{joints}} |\ddot{q}_j|^2$ | -2e-6 |
| Weighted Torques | $\sum_{j \in \mathrm{joints}} |\tau_j/\mathrm{kp}_j|^2$ | -1e-7 |
| Contact Forces | $\mathbf{1}\{|F_i| >= F_{\mathrm{th}}\} * \{|F_i| - F_{\mathrm{th}}\}$ | -3e-4 |
| Collision | $\sum_{i \in \mathrm{contact}} \mathbf{1}\|F_i\| > 0.1$ | -10. |
| Action Rate | $\sum_{j \in \mathrm{joints}} |a_{t-1} - a_t|^2$ | -6e-3 |
| Arm Dof Err | $\sum_{j \in \mathrm{arm\ joints}} |q_j|^2$ | -0.3 |
| Waist Dof Err | $\sum_{j \in \mathrm{waist\ joints}} |q_j|^2$ | -0.1 |
| Hip Yaw Dof Err | $\sum_{j \in \mathrm{hip\ yaw\ joints}} |q_j|^2$ | -0.1 |
| Feet Away | $\min(\|p_{\mathrm{left\ foot}} - p_{\mathrm{right\ foot}}\|, 0.4)$ | 0.4 |

$g = (g_x, g_y, g_z)$ is the projected gravity vector, which is the gravity vector in the robot base frame. To protect the hardware motor, we add a penalty to the motor torques. However, not all joints have the same motor capacity and the same kp factor. We weigh the torque with the kp factor of each joint so that each motor joint can have a relatively equal penalty. To be noted that, none of these reward terms involve motion reference for the humanoid robot's locomotion.

Table 5: PPO Parameters

| | |
|---|---|
| PPO clip range | 0.2 |
| GAE $\lambda$ | 0.95 |
| Learning rate | 3e-5 |
| Reward discount factor | 0.99 |
| Minimum policy std | 0.2 |
| Number of environments | 4096 |
| Number of environment steps per training batch | 24 |
| Learning epochs per training batch | 5 |
| Number of mini-batches per training batch | 4 |

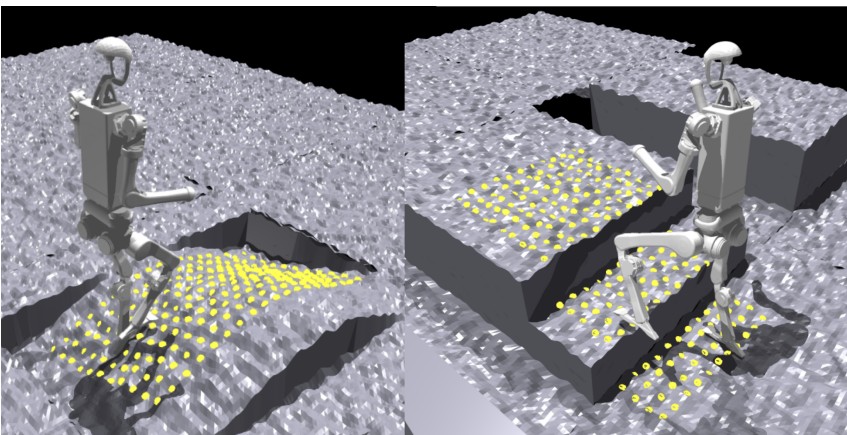

Figure 8: Scandots and Fractal Noise Terrain example. In this figure, the yellow dots show how scandots are visualized in wave terrain and discrete terrain.

## D    Simulation terrain details

In Table 6, we list out the critical attribute of each type of parkour obstacle.

## E    Policy Architecture

The oracle parkour policy consists of an RNN-MLP state estimator, an RNN-MLP actor, and an MLP terrain encoder. The student parkour policy consists of the same architecture in the RNN-MLP state estimator and the RNN-MLP actor. The student parkour policy uses a CNN network for the depth encoder. The detailed parameters of the network structure are listed in Table 7.

Table 6: The training and testing ranges of the critical properties of each parkour obstacle.

| Parameter | Training Range | Testing Range |
|---|---|---|
| Jump Height (m) | [0.2, 0.5] | [0.2, 0.6] |
| Down Height (m) | [0.1, 0.6] | [0.2, 0.6] |
| Leap Length (m) | [0.2, 1.2] | [0.2, 1.2] |
| Slope Angle (rad) | [0.2, 0.42] | [0.2, 0.4] |
| Stairs Height (m) | [0.1, 0.3] | [0.1, 0.3] |
| Stairs Length (m) | [0.3, 0.5] | [0.3, 0.5] |
| Hurdle Height (m) | [0.05, 0.5] | [0.1, 0.5] |
| Ramp Angle (rad | [0.2, 0.5] | [0.2, 0.4] |

Table 7: Parkour Policy Structure

| | | |
|---|---|---|
| Actor | RNN type | GRU |
| | RNN layers | 1 |
| | RNN hidden dims | 256 |
| | MLP hidden sizes | 512, 256, 128 |
| | MLP activateion | CELU |
| Scandot Encoder | MLP hidden sizes | 128, 64 |
| | MLP activation | CELU |
| | Encoder embedding dims | 32 |
| Depth Encoder | CNN channels | 16, 32, 32 |
| | CNN kernel sizes | 5, 4, 3 |
| | CNN polling layer | MaxPool |
| | CNN stride | 2, 2, 1 |
| | CNN embedding dims | 32 |

Table 8: The $kp$ $kd$ factors and maximum torques for each joints

| Joint Names | $kp$ | $kd$ | Torque Limits (Nm) |
|---|---|---|---|
| shoulder pitch | 30 | 1. | 40 |
| shoulder roll | 30 | 1. | 40 |
| shoulder yaw | 20 | 0.5 | 18 |
| elbow | 20 | 0.5 | 18 |
| torso | 200 | 3 | 200 |
| hip yaw | 60 | 1.5 | 200 |
| hip roll | 220 | 4 | 200 |
| hip pitch | 220 | 4 | 200 |
| knee | 320 | 4 | 300 |
| ankle | 40 | 2 | 40 |

## F Deployment Details

The depth images in both the simulation and the real robot are acquired from a $480 \times 640$ sensor resolution. Then, the depth image is processed and resized to $48 \times 64$ resolution. The proprioception is sent by the motor system through Cyclone DDS at 500Hz. We run the visual encoder process at 10Hz and the rest of the parkour network at 50Hz. The vision encoder process acquires a depth image, computes the image latent, and sends the latent vector to the parkour network using ROS message. The parkour network sends joint position targets to the motors with different $kp$ and $kd$ factors on each joint, shown in Table 8. The motors generate torque through their built-in PD controller. To ensure safety, we clipped the joints target position based on the torque limit $\tau_{\max}$: $\text{clip}\left(q^{\text{target}}, (kd * \dot{q} - \tau_{\max})/kp + q, (kd * \dot{q} + \tau_{\max})/kp + q\right)$

We use the Unitree H1 humanoid robot for our real-world test, which is equipped with an Intel RealSense D435i and an Intel Core i7 12-core NUC onboard. The robot has 19 joints including the arm and waist. We assign different $kp$, $kd$ parameters for each joint as shown in Table 8. We use ROS2 and Cyclone DDS for the communication between the policy and motors. The depth encoder sends its embedding to the recurrent network at 10Hz. The recurrent network receives and sends the target positions of all joints to the motors at 50 Hz. The motors compute the torque based on their built-in PD controller at 1000Hz.

# G    Further Results Compared with feet_air_time reward

| | Success Rate (%) ↑ | | | Average Distance (m) ↑ | | |
|---|---|---|---|---|---|---|
| | Jump Down | Slope | Stairs Down | Jump Down | Slope | Stairs Down |
| feet_air_time | 100 | 100 | 100 | 14.4 | 14.4 | 14.4 |
| Fractal (Ours) | 100 | 100 | 100 | 14.4 | 14.4 | 14.4 |

Table 9: We compare the success rate and average moving distance in simulation. Each type of terrain is a connected 3 sub-track with linearly increasing difficulty. Each sub-track has a length of 4.8m, which makes the total length of a given terrain 14.4m.

| | Success Rate (%) ↑ | | | Average Distance (m) ↑ | | |
|---|---|---|---|---|---|---|
| | Discrete | Wave | Tilted Ramp | Discrete | Wave | Tilted Ramp |
| feet_air_time | 100 | 30 | 100 | 14.4 | 11.6 | 14.4 |
| Fractal (Ours) | 100 | **100** | 100 | 14.4 | **14.4** | 14.4 |

Table 10: We compare the success rate and average moving distance in simulation. Each type of terrain is a connected 3 sub-track with linearly increasing difficulty. Each sub-track has a length of 4.8m, which makes the total length of a given terrain 14.4m.

# H    Results Compared with Different Distillation Methods

| | Success Rate (%) ↑ | | | | Average Distance (m) ↑ | | | |
|---|---|---|---|---|---|---|---|---|
| | Jump up | Leap | Stairs up | Hurdle | Jump up | Leap | Stairs up | Hurdle |
| From Scratch | 0 | 0 | 5 | 10 | 2.6 | 2.8 | 3.4 | 7.3 |
| With one GPU | 40 | 45 | 65 | 25 | 8.6 | 10.2 | 9.8 | 7.2 |
| Parkour (Ours) | **85** | **80** | **100** | **95** | **13.8** | **14.0** | **14.4** | **14.1** |

Table 11: We compare the success rate and average moving distance in simulation. All three methods use the same oracle policy as the teacher policy and run 24 hours for the results. Distillation using only a Single GPU or distilling from a randomly initialized student policy results in worse performance. (More experiment results for the other obstacles are shown in the appendix.)

Compared with previous work using quadruped robot parkour [3, 4], we aim to investigate and answer why multi-GPU acceleration is necessary to distill such a humanoid parkour policy. We use the same oracle policy and distill the student policy under 3 different settings, shown in Table 11 12 13. We run 20 tails for each setting in simulation. We distill the student parkour policy using a randomly initialized network. After 24 hours of distillation, the student policy still struggles to walk steadily. We test the distillation from the oracle parkour policy but only one GPU process to collect the labeled trajectory and train the policy. After 24 hours, it only trains on $4.147 \times 10^6$ transitions compared to $432 \times 10^6$ transitions in 4-GPUs variants. This one-GPU variation needs more time to perform as well as our student parkour policy. The one-GPU variant is aware of different types of obstacles, but the legs do not raise high enough to overcome these obstacles.

|  | Success Rate (%) ↑ | | | Average Distance (m) ↑ | | |
|---|---|---|---|---|---|---|
|  | Jump Down | Slope | Stairs Down | Jump Down | Slope | Stairs Down |
| From Scratch | 15 | 20 | 0 | 7.6 | 5.1 | 2.4 |
| With one GPU | 80 | 70 | 65 | 13.2 | 9.8 | 9.2 |
| Ours (Parkour) | **100** | **100** | **100** | **14.4** | **14.4** | **14.4** |

Table 12: We compare the success rate and average moving distance in simulation. All three methods use the same oracle policy as the teacher policy and run 24 hours for the results. Distillation using only a Single GPU or distilling from a randomly initialized student policy results in worse performance.

|  | Success Rate (%) ↑ | | | Average Distance (m) ↑ | | |
|---|---|---|---|---|---|---|
|  | Discrete | Wave | Tilted Ramp | Discrete | Wave | Tilted Ramp |
| From Scratch | 0 | 10 | 5 | 2.3 | 6.7 | 5.5 |
| With one GPU | 75 | 85 | 85 | 11.6 | 13.1 | 12.9 |
| Ours (Parkour) | 100 | **100** | **100** | **14.4** | **14.4** | **14.4** |

Table 13: We compare the success rate and average moving distance in simulation. All three methods use the same oracle policy as the teacher policy and run 24 hours for the results. Distillation using only a Single GPU or distilling from a randomly initialized student policy results in worse performance.

