# OpenReview forum: "Humanoid Parkour Learning"
_robot-learning.org/CoRL/2024/Conference — CoRL 2024_

### Official Review · Reviewer_XFVA · 2024-07-19

**Originality:** 4
**Technical Quality:** 4
**Clarity Of Presentation:** 4
**Potential Impact:** 3
**Recommendation:** 3
**Confidence:** 4

**Review:**

**Strengths:**

1. The paper introduces a novel framework for training humanoid robots to perform parkour skills using fractal noise in terrain, which is a creative solution to the challenge of locomotion without motion priors.

2. The authors provide a detailed description of their training method, including the use of scandots for terrain perception and DAgger for policy distillation.

3. The successful demonstration of sim-to-real transfer is significant, showing that the policies trained in simulation can be effectively deployed in real-world scenarios.

4. The framework is very robust in the real world, and can even balance when the upper body is teleoperated.

**Weaknesses:**

1. Though I appreciate the efforts in engineering, the overall technical novelty is limited. The framework to do robot parkour has been well proposed in previous works such as Robot Parkour Learning and Extreme Parkour.

2. The pure RL pipeline might make the parkour motions of humanoid robots not so natural, compared to motion-based methods.

3. I would suggest a title change to make the paper more accurate and not overclaimed. I fully appreciate the success of making humanoid robots walk in challenging terrain in real world, while I do think the current demo shown by authors is not enough to cover "Humanoid Parkour", which is an ambitious goal.

**Quality Of The Limitations Section:**

3

**Questions For Rebuttal:**

See weaknesses.

**Robotics Focus:**

4

**Summary Of Paper:**

This paper proposes a sim-to-real pipeline that enables humanoid robots to successfully parkour in the real world.

**Summary Of Recommendation:**

This paper presents high quality real world experiments, and thus I recommend an accept. However, the current title seems to be overclaiming. I would recommend a title change before acceptance to be rigorous.

---

### Official Review · Reviewer_CnFz · 2024-07-20
**Review for Humanoid Parkour Learning**

**Originality:** 2
**Technical Quality:** 3
**Clarity Of Presentation:** 3
**Potential Impact:** 3
**Recommendation:** 3
**Confidence:** 5

**Review:**

**Strengths**
- Very impressive humanoid parkour demos in the real world.
- The paper is easy to follow and the presentation is clear.
- Training these parkour policies is not easy, and it is really nice to these sim2real tranfer works for humanoid parkour.

**Weaknesses**
- Methodology wise, I don't see a big difference of this paper and previous quadruped parkour works.
- Some claims in paragraph 3 of introduction are not well-justified:
    - why diverse skills for a single locomotion network are challenging? Many works have already proven diverse skills for a single locomotion network.
    - why mimicking animals or humans lacks egocentric perception? Adding perception is definitely doable to train vision-based mimicking policy. It is an option to add on, and mimicking-style work could scale to perception. I don't see why "This significantly restricts the scalability of these methods."
- Fractal noise needs more detailed introduction since the paper claims several times the advantage of fractal noise over legged reward tuning.
- Why use straight parkour lanes for training instead of omni-directional terrains? This could saves the difficulty to handle "Auto Command to Walk along the Track"
- Also, for the oracle parkour policy training, each column is a single type of obstacle. Does this mean during training, the humanoid is trained to transverse over a lane with same obstables? I wonder how the humanoid performs when there is a lane with various terrain types ahead. This experiment is needed
- The following claim seems not right "Concurrent works use “feet air time” [21, 23] or motion reference to make the walking policy able to walk up and down stairs without vision". However, these two metioned works do not show any stairs walking demo to the best of my knowledge.
- The caption and content of Table 2 is too close.
- The paper reports a lot of success rates but the number of runs are missing, both in real world and simulation.
- This paper needs more experimental results to justify the design choices.
    - For example, how does GRU compare with other more recently used architecures like Transformer or simple MLPs with short-term histories.
    - Curves in Figure 5 does not show the limit of the proposed learning method. When will the proposed method fail? Is it a hardware limit or training design limit?
    - Some abaltions are needed, for example, why types of 10 terrains? Is 10 enough of not enough? What is the performance of parkour policy with only 2/3/5/7 terrains types? How much is the proposed method gonna benefit from more terrains?

**Quality Of The Limitations Section:**

3

**Questions For Rebuttal:**

The questions are summarized in the weakness section. I am happy to increase my scores if my concerns could be answered.

**Robotics Focus:**

4

**Summary Of Paper:**

This paper showcases how to leverage the recent succuss of quadruped parkour learning and apply it to a much harder problem - humanoids. The real world demos are impressive and convicing, but the paper writing and experimental results need more justifications.

**Summary Of Recommendation:**

The real-world demo is impressive, but the paper needs more detailed writing and more experiments to justify the design choices. I vote for weak reject for the current paper presentation.  But I would be happy to raise my scores if my concerns could be addresed.

---

### Official Review · Reviewer_sc8E · 2024-07-21
**Strong real world results on an important problem**

**Originality:** 3
**Technical Quality:** 4
**Clarity Of Presentation:** 4
**Potential Impact:** 3
**Recommendation:** 4
**Confidence:** 4

**Review:**

- The paper is very well written, clear and easy to follow. The method figures and the tables are also quite clear
- The paper employs a 3 stage simulation based RL training to first train a velocity tracking policy, followed by a teacher parkour policy that uses privileged information and then finally distills the teacher policy into the student policy. The high level pipeline is a common approach to learned locomotion and the paper extends that to achieve sim to real transfer of complex behaviors for bipeds which poses difficulty in behavior learning, as well as sim to real transfer since the bar is much higher for both.
- The paper will have an impact is demonstrating the feasibility of learning complex maneuvers in sim and then transferring them zero-shot to real.
- The paper over looks strong, modulo a few ablation experiments which are detailed below.
- The section on "upper limb control" is a very impressive side application of the proposed method.
- A more detailed limitations discussion would help the paper. For example, the best humanoid parkour till date is still the ones shown by boston dynamics. However, the limitation section is limited to talking about manipulation as the next missing piece without addressing the limited behaviors achieved in parkour.

**Quality Of The Limitations Section:**

2

**Questions For Rebuttal:**

- The paper proposes 3 learning phases. Is there an ablation where the learning in the first two phases is combined together to understand the extent to which it affects performance?
- An analysis of sim to real gap on high speed or agile maneuvers is missing. I would expect the performance of sim to real to be worse for fast and agile motions than slow motions. It would help to have an understanding of that in any simple way possible.
- It is not clear if the paper maxes out the limits of the hardware in terms of jumping, etc. It would be good to understand to what limit is the hardware used of its full capacity.
- The limitations section should additionally discuss whats missing to achieve the same level of agility as achieved by Boston Dynamics via the use of learning based method.

**Robotics Focus:**

4

**Summary Of Paper:**

The paper proposes a sim to real RL based method for parkour on a humanoid robot (H1) in the real world using egocentric depth. The method is gives very strong real world results.

**Summary Of Recommendation:**

The paper pushes the frontier on sim-to-real RL for parkour on humanoid robots. Modulo a few clarification questions, the paper looks quite strong.

---

### Decision · Program_Chairs · 2024-09-04

**Decision:**

Accept

**Comment:**

This paper introduced a sim-to-real RL framework for learning vision-based humanoid locomotion behaviors using egocentric depth.  It utilizes a three-stage RL training process to achieve agile parkour behaviors and successfully deploys the learned controller to real robot hardware (Unitree H1). This work received mixed initial reviews, with one strong accept, one weak reject, and one weak accept. The reviewers pointed out the work's strengths as 1) the real robot results demonstrated convincing evidence of the effectiveness of this method, 2) the overall presentation of the work is good, and the detailed description of the training process is clear, and 3) they demonstrated the robustness of the controller while teleoperating the humanoid upper body. In the meantime, some major weaknesses were also mentioned by the reviewers, including 1) the work lacks substantial technical innovations, especially considering prior work on quadruped locomotion, 2) additional ablation studies and analyses (e.g., separation of the three training phases, the impact of training on diverse terrains, performance limits of the proposed method) are needed to improve the quality of the draft, and 3) the demonstrated robot behaviors, while impressive, may not fully encompass the ambitious goal of "humanoid Parkour," so a more accurate title that avoids overclaiming might be considered.

**Post-rebuttal update:**

The AC appreciated the detailed responses and additional experiment results in the authors' rebuttal, which definitely improved the quality of the submission. Toward the end of the discussion period, Reviewer CnFz upgraded their rating to Weak Accept, making the three votes to be unanimously positive. The AC believes that, with the positive changes during the rebuttal phrase, this paper has passed the threshold of publication for CoRL and would recommend acceptance of this work.